# Logic Extraction: Enhancing AI Generalization in Abstraction and Reasoning Corpus Tasks

**Shaoting Zhu**\*
2024311565

**Shuangyue Geng**\*
2024311543

**Un Lok Chen**\*
2024270027

## 1 Introduction

The concept of Artificial General Intelligence (AGI) has diffused rapidly to the public after the success of ChatGPT[1], but the quest for *what is intelligence?* has a much longer history. In 2019, François Chollet published a paper called "On the Measure of Intelligence," in which he provided an in-depth discussion of how we should define and measure intelligence in terms of the efficiency of *skill acquisition*, rather than the level of skills[2]. Along with the paper, a new benchmark called the Abstraction and Reasoning Corpus (ARC) dataset was released.

### 1.1 The ARC Benchmark

Through the lens of knowledge priors, experience (an agent/system's exposure to the new task), and generalization difficulty (of a task), Chollet carefully designed a series of pattern-finding geometric tasks in the ARC dataset[2]. As an example illustrated in Fig. 1, the tester should infer the output grid of a testing input after observing a few input-output pair examples.

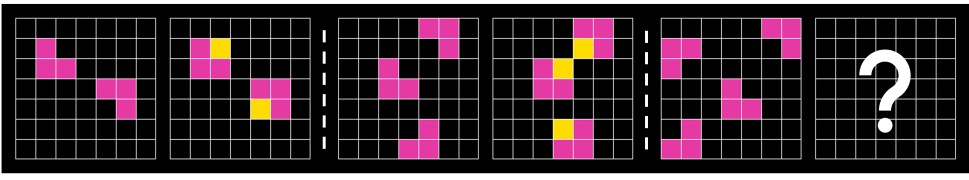



Example 1 (Input-Output)      Example 2 (Input-Output)      Test (Input-Output)



Figure 1: An example of the ARC geometric task.

The intelligent system is supposed to possess the following Core Knowledge priors that are relatively innate to us humans: objectness, goal-directedness, numbers and counting, and basic geometry and topology[2]. The generalization difficulty of the tasks is guaranteed by the combinations and transformations of concepts in these 4 domains, while experience is explicitly controlled by the limited number of examples in each task.

### 1.2 Challenges and Impacts

The average human performance on the ARC public test is above 60% accuracy[3]. On the contrary, the most competent models can only achieve accuracy below 50% leveraging SOTA LLMs[4]. The gap between current AI and humans is more distinct when considering the vast amount of pre-training data. The investigation of solutions to the ARC competition could bring us significant insights into modeling both the intuition and reasoning process in the human mind, promoting the construction of novel AI paradigms. Meanwhile, "[a]t minimum, solving ARC-AGI would result in a new programming paradigm[5]," enabling program synthesis for people who do not have experience in coding by simply showing a few input-output examples.

## 2 Competition Specifics

**Dataset**    The ARC Prize competition provides three datasets: public training set, public evaluation set, and private evaluation set. Both the public training and public evaluation sets include 400 task files, while the private evaluation set consists of 100 task files. Each task has 2 to 10 pairs (typically 3) of examples and 1 to 3 pairs (typically 1) of tests[2, 6].

**Metrics**    We can assess performance through two approaches: 1) Pixel correctness–the percentage of pixels correctly inferred out of the total; 2) Correct / Incorrect–whether or not the inferred output matches the task's test output in terms of shape, color, and position. The competition evaluates submissions using the second approach[6].

---

\*Equal contribution (authors listed in alphabetical order).

Preprint. Course project.

# 3 Related work

**SOTA Methods in ARC Task**   There are two mainstream approaches to tackle the ARC task: Domain Specific Language (DSL) Program Synthesis and Active Inference. Both winners of ARC 2020 and 2022 adopted the former method[7, 8]. Typically, the DSL primitives are manually defined and would encode basic concepts such as "objects" and implement functions such as "rotate". The solver may be a searching program on the compositional space of the primitives, encompassing simple brute force searching to more advanced genetic algorithms[9, 10, 11]. Although direct prompting and fine-tuning on LLMs are ineffective in the ARC challenge[6], active inference with LLMs championed the ARC 2023 competition and has boosted performance to 49% in 2024[4]. The winner of ARC 2023 developed a framework called AIRV (augment-inference-reverse augmentation-vote) that performs test-time fine-tuning and augmentation on synthetic data[12].

**Information Bottleneck**   The Information Bottleneck (IB) approach provides a balance between data compression and retention of essential information, enhancing its predictive or representational capabilities[13]. IB has been successfully applied in various deep neural network tasks, such as in the natural language processing[14, 15, 16] and computer vision[17, 18] domain. In deep neural networks, IB typically facilitates a direct mapping from the input data to the desired output, thereby enabling the learning of robust feature representations.

# 4 Proposed Method

We propose a framework that mimics the cognitive process humans employ when solving ARC tasks. Particularly, humans can generalize the rules or patterns observed in the images into a higher-level "inner logic". In addition, this inner logic is often more abstract than explicit expressions like natural language or code. We aim for our model to autonomously learn this implicit representation of the inner logic as an **information bottleneck**. Our proposed method involves the following three steps.

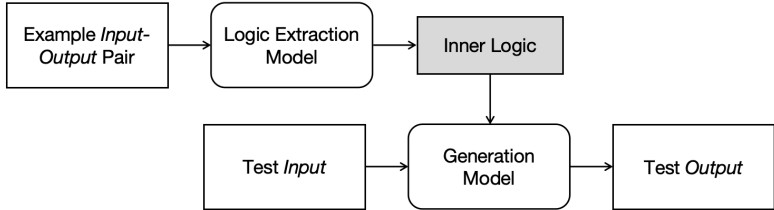

Figure 2: The basic pipeline of our proposed method, using inner logic as an information bottleneck to connect two models.

## 4.1 Data augmentation

We plan to implement data augmentation in two dimensions: 1) increasing the number of training examples for each task and 2) expanding the feature dimension of the data with explicit annotations of the inner logic (e.g. natural language or Domain Specific Language(DSL)). For the former, additional data will be generated through techniques like recoloring, flipping, rotating, and other program-based methods[19, 20]. For the latter, logic annotations will be produced using an agentic system within a closed-loop framework, referring to [21].

## 4.2 Basic pipeline

As shown in Fig. 2, our pipeline consists of two models: a logic extraction model and a generation model. The logic extraction model is trained to distil the inner logic from example input-output pairs. Possible architectures for this model include LLMs, CNNs, etc. The generation model applies the extracted logic to the input to generate the desired output. It could take the form of a rule-based program, an image-based neural generative network (such as CNNs or diffusion models), or an autoregressive matrix generative network. Since supervision data is available for both models (from sections 4.1), they can be trained independently. Curriculum training method will be introduced to improve training stability. We can divide the dataset into several different levels of difficulty according to their logical complexity, and start model training with simple tasks first.

## 4.3 Emergent language

We aim to represent the inner logic using an implicit emergent language learned by the model itself, rather than relying on explicit language-based or rule-based logic. This is achieved by jointly training two models after completing the initial training described in section 4.2. Furthermore, we may introduce an additional decoder head to translate it back into explicit representations, using a reconstruction loss to guide the training.

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
