# OpenReview forum: "【Proposal】Logic Extraction: Enhancing AI Generalization in Abstraction and Reasoning Corpus Tasks"
_tsinghua.edu.cn/THU/2024/Fall/AML — THU 2024 Fall AML Submission_

### Official Review · ~王俊逸1 · 2024-11-08
**Advancing AI's Abstract Reasoning through Logic Extraction**

**Rating:** 8
**Confidence:** 4

**Review:**

This proposal, "Logic Extraction: Enhancing AI Generalization in Abstraction and Reasoning Corpus Tasks," takes on the formidable challenge of bolstering AI's capacity for abstraction and reasoning as measured by the Abstraction and Reasoning Corpus (ARC) benchmark. The authors' approach is commendable for its ambition to bridge the performance gap between human intuition and current AI capabilities on complex pattern recognition tasks.

The framework's cornerstone is the extraction and application of an "inner logic" that mirrors human problem-solving, which is both innovative and aligned with the cutting-edge of AI research. The proposed method's three-pronged attack—data augmentation, a dual-model pipeline, and the emergence of an implicit language—is a strategic move that could potentially revolutionize how AI systems tackle ARC tasks.

Yet, the proposal's success hinges on the effective development and translation of the emergent language, a component that requires further elucidation. The practical considerations, including computational demands and the nuances of training data, are also critical factors that deserve deeper exploration to assess the feasibility of this ambitious project.

Overall, the proposal's vision to enhance AI's generalization through logic extraction is both laudable and timely. It holds the promise of not only improving AI performance on the ARC but also contributing to the broader quest for Artificial General Intelligence. The potential of this work to shed light on the human reasoning process and to facilitate a new era of programming by example is undeniable, marking it as a significant endeavor in the field of AI.

---

### Official Review · ~Liu_Yating1 · 2024-11-08
**Well done**

**Rating:** 9
**Confidence:** 4

**Review:**

The proposal presents a novel framework that aims to enhance AI's generalization capabilities in abstract reasoning tasks by mimicking human cognitive processes. The proposed method's reliance on information bottleneck theory to learn implicit representations of inner logic is a theoretically sound approach that aligns with current trends in deep learning and feature representation learning. However, a general framework is recommended to be proposed in Background to help readers better understand your work.

---

### Official Review · ~Guangjie_Xu1 · 2024-11-09

**Rating:** 9
**Confidence:** 4

**Review:**

The ARC task is a pioneering benchmark designed to test AI’s generalization, abstract reasoning, and pattern recognition, closely mimicking aspects of human cognition. By requiring minimal data for each task, it challenges models to understand concepts rather than memorize patterns, setting a higher bar for Artificial General Intelligence (AGI). This approach could inspire new AI paradigms that integrate intuition with structured reasoning, essential for complex, real-world problem-solving.

**Pros**:
1. Encourages True Generalization: ARC pushes models to rely on conceptual understanding, fostering skills beyond pattern memorization.
2. Human-Like Reasoning Framework: Emphasizes core knowledge such as object recognition, goal orientation, and basic spatial reasoning, aligning AI tasks more closely with human learning.
3. Promotes Minimal Data Usage: Limits data dependence, creating a meaningful challenge for models to learn through fewer examples and highlighting their cognitive adaptability.

**Cons**:
1. Difficult for Current AI Models: Present AI systems achieve low accuracy, revealing a significant gap between human and machine reasoning.
2. Abstract Nature May Limit Practicality: The tasks are highly abstract, which may make them less representative of real-world applications, limiting their immediate usability.
3. Risk of Overfitting to Specific Benchmarks: Focusing solely on ARC could encourage models to adapt to ARC-specific quirks rather than develop true general reasoning abilities.

---

### Official Review · ~Xiaoqian_Liu7 · 2024-11-10
**Great Work**

**Rating:** 9
**Confidence:** 3

**Review:**

The paper presents a compelling approach to improving AI's ability to generalize and reason abstractly, as measured by the Abstraction and Reasoning Corpus (ARC) benchmark.

The authors have identified a significant gap in current AI capabilities compared to human performance on the ARC and propose a novel framework that leverages data augmentation, a dual-model pipeline, and the emergence of an implicit language to bridge this gap.

---

### Official Review · ~Keyu_Shen1 · 2024-11-10
**Insightful and Ambitious Proposal**

**Rating:** 9
**Confidence:** 3

**Review:**

The proposal presents a well-structured and ambitious approach to enhancing AI’s generalization capabilities in Abstraction and Reasoning Corpus (ARC) tasks. It effectively outlines the current limitations of AI in achieving human-level performance in these tasks and highlights the need for models capable of abstract reasoning. By introducing innovative methods such as emergent language and logic extraction, the proposal shows significant potential. However, it would benefit from further discussion on computational efficiency and practical feasibility, especially given the complexity of the dual-model framework.

---

### Official Review · ~Zhixuan_Pan1 · 2024-11-11

**Rating:** 7
**Confidence:** 3

**Review:**

This project aims to improve artificial intelligence's generalization abilities in Abstraction and Reasoning Corpus (ARC) tasks by emulating human cognitive processes to extract underlying logic. It leverages the Information Bottleneck (IB) method as an intermediary, linking a logic extraction model with a generation model.

Pros:

1. The proposal’s objectives are highly specific, and the methods presented are concrete and detailed.

2. The ARC benchmark is challenging, and achieving strong performance in this competition would be meaningful with practical significance. Additionally, it has the potential to establish a new paradigm for reasoning.

Cons:

1. The relationship between the logic extraction model and the generation model is not specified. Given that they work in parallel to form an overall model for ARC task, I believe they should not train independently. If they are truly trained separately, which part would be considered the primary intelligent component? Additionally, I speculate that the ARC competition imposes constraints on model size, but the proposal does not address how the model size allocation between these two models is managed.

2. Why not use a generation model with Chain of Thought (CoT) prompting as the logic extraction model?

3. Emergence is typically a phenomenon observed in large-scale language models, and I am uncertain whether the model you plan to train will exhibit emergent capabilities.

---

### Official Review · ~Changsong_Lei2 · 2024-11-11
**review of "Logic Extraction: Enhancing AI Generalization in Abstraction and Reasoning Corpus Tasks"**

**Rating:** 8
**Confidence:** 4

**Review:**

### Summary:
The proposal discusses using a logic extraction framework to improve AI generalization capabilities on the Abstraction and Reasoning Corpus (ARC), a benchmark for tasks that require abstract reasoning and pattern recognition.

### Pros:
- The proposal introduces a good approach of using inner logic as an information bottleneck, aligning well with the abstract reasoning requirements of ARC tasks.
- The proposed pipeline is clear, where one model extracts logic and the other generates outputs, is logically organized and offers flexibility in choosing suitable architectures.
- The proposal’s plan to create an implicit, emergent language for rule representation is ambitious and could add interpretability to the model’s reasoning process.

### Cons:
- The proposal’s idea of an "inner logic" and emergent language could be difficult to quantify. Specific metrics for measuring these abstract representations would improve clarity.

Generally speaking, this proposal gives an innovative and thoughtful approach to replicating human-like abstract reasoning in AI, it is well written.

---

### Official Review · ~Yu_Zhang61 · 2024-11-11
**Review of "Logic Extraction: Enhancing AI Generalization in Abstraction and Reasoning Corpus Tasks"**

**Rating:** 8
**Confidence:** 4

**Review:**

This thesis proposal introduces a framework aimed at advancing artificial intelligence's capacity for generalization on the Abstraction and Reasoning Corpus (ARC) benchmark by extracting implicit "inner logic" from examples and using it as an information bottleneck to promote more human-like cognitive processing in pattern recognition and reasoning tasks. This approach leverages data augmentation, curriculum training, and emergent language to strengthen AI's abstraction capabilities. The proposal is commendable for its originality and relevance to AI generalization, as it addresses a critical gap between AI and human cognitive performance. However, the proposed methodology would benefit from greater detail on the operationalization and validation of "inner logic" extraction, as this process could be complex and challenging to evaluate. Additionally, the computational requirements of the proposed methods are not addressed, potentially limiting the project’s feasibility. Overall, while promising, the proposal would benefit from further clarity in methodology and implementation details to ensure practical viability and impact.

---

### Official Review · ~Jiaxiang_Liu7 · 2024-11-11
**Innovative Approach with Potential**

**Rating:** 8
**Confidence:** 4

**Review:**

This paper presents an innovative framework aimed at enhancing AI generalization within the Abstraction and Reasoning Corpus (ARC) tasks by replicating human cognitive processes. The approach leverages a logic extraction model to distill inner logic as an information bottleneck, coupled with a generative model for output production. Notable elements include data augmentation strategies, curriculum training to boost stability, and a distinctive emergent language approach for implicit logic representation. Overall, this work is creative and addresses critical challenges in AI generalization.

---

### Official Review · ~Zhaoxi_Li2 · 2024-11-12
**Proposal Review: Enhancing AI Generalization in Abstraction and Reasoning Corpus (ARC) Tasks**

**Rating:** 10
**Confidence:** 4

**Review:**

This proposal outlines an innovative approach to enhancing AI generalization in the Abstraction and Reasoning Corpus (ARC) tasks by developing a model that learns implicit "inner logic" as an information bottleneck, mirroring human cognitive processes. By leveraging data augmentation, a two-model pipeline for logic extraction and output generation, and a unique emergent language representation, the authors address limitations in current ARC task approaches, which struggle with generalization and pattern recognition. The proposal is well-structured and grounded in relevant literature, detailing an efficient framework with potential for high-impact advancements in AI reasoning capabilities. However, providing further details on evaluation metrics and comparative studies would strengthen the assessment of model performance and generalization in future work.

---

### Official Review · ~Xuancheng_Li1 · 2024-11-12

**Rating:** 9
**Confidence:** 4

**Review:**

Summary
This proposal introduces a framework aimed at improving AI’s generalization on the ARC benchmark by mimicking human-like reasoning. Through logic extraction, an information bottleneck, and data augmentation, the approach seeks to capture underlying patterns in ARC tasks, enhancing robustness and abstraction capabilities in AI models.

Strengths
The framework is innovative, leveraging concepts like emergent language and curriculum training to address gaps in AI’s reasoning and pattern recognition, which could advance AI's generalization capabilities significantly.

Weaknesses
The proposal would benefit from more concrete examples of the “inner logic” extraction process and specific performance metrics for evaluating improvements.

Conclusion
This research offers a promising direction for enhancing AI reasoning on challenging tasks, with potential implications for advancing general AI capabilities. Further work could clarify the practical applications of the proposed framework and provide detailed evaluations.

---

### Official Review · ~Yifan_Luo2 · 2024-11-12
**Good Problem for Nowadays LLMs**

**Rating:** 9
**Confidence:** 5

**Review:**

### Summary

The proposal discusses enhancing AI generalization in Abstraction and Reasoning Corpus (ARC) tasks by using logic extraction. The authors propose a framework that mimics human cognitive processes, aiming to autonomously learn implicit "inner logic" through an information bottleneck.

### Pros

1. **Innovative Approach**: The use of an information bottleneck to extract and apply inner logic is a novel method that could significantly improve AI generalization.
2. **Human-Centric Design**: Mimicking human cognitive processes could bridge the gap between AI and human performance.

### Cons

1. **Complexity**: Implementing and training this system may require substantial computational resources and time.
2. **Uncertainty in Emergent Language**: The reliance on an implicit emergent language might pose challenges in interpretability and debugging the model.